# Quantum Entanglement Trees: Optimizing Quantized Matrix Quantization via Element Replacement and Residual Clustering

## Abstract

The matrix quantization entails representing matrix elements in a more space-efficient form to reduce storage usage, with dequantization restoring the original matrix for use. We formulate the Quantization Error Minimization (QEM) problem as minimizing the distance between a matrix before and after quantization, under the condition that the quantized matrix occupies the same memory space. Matrix quantization is crucial in various applications, including Large Language Models (LLMs) weight quantization, vector databases, KV cache quantization, graph compression, and image compression. Recent advancements in LLMs, such as GPT-4 and BERT, have highlighted the importance of matrix compression due to the large size of parameters and KV cache, which are stored as matrices.

We propose Quantum Entanglement Trees (QET) to address the QEM problem by leveraging the local orderliness of matrix elements, involving iterative element swapping to form a locally ordered matrix. This matrix is then grouped and quantized by columns. To enhance QET, we introduce two optimizations: Residual Quantization Optimization (RQO), which reduces MSE by quantizing the residuals between the original and dequantized matrices, and Codebook Quantization Optimization (CQO), which reduces storage requirements by compressing the codebook itself.

Experimental results demonstrate that QET can effectively reduce MSE to 5.05%, 13.33%, and 11.89% of the current best method on the LLM dataset, K cache, and V cache, respectively. Our contributions include the abstraction of the QEM problem, the design of the QET algorithm, and the proposal of two optimizations to improve accuracy and speed.

## 1 Introduction

The matrix quantization entails representing the elements of a matrix in a more space-efficient form to reduce its storage usage. Dequantization, on the other hand, is the process during usage where the original matrix is restored from the quantized matrix using a restoration algorithm. We formulate the Quantization Error Minimization (QEM) problem as the task of minimizing the distance between a matrix before and after quantization in high-dimensional space, under the condition that the quantized matrix occupies the same space.

Matrix quantization is widely employed across diverse applications, including Large Language Models (LLMs) weight quantization (Dettmers et al. (2024); Lin et al. (2024); Shao et al. (2023); Xiao et al. (2023)), vector database (Xu et al. (2018); Jegou et al. (2010)), LLM k-v cache quantization (Liu et al. (2024); Zhang et al. (2024); Hooper et al. (2024); Kawakibi Zuhri et al. (2024); Duanmu et al. (2024); Yue et al. (2024); Lee et al. (2024); Adnan et al. (2024)), graph compression (Brisaboa et al. (2009); Claude & Ladra (2011)), and image compression (Yu et al. (2018); Ning et al. (2016)).

Specifically, the recent advancements in Large Language Models (LLMs) have made matrix compression even more critical. Large Language Models (LLMs) have revolutionized the field of natural language processing (NLP), enabling significant advancements in tasks such as machine translation, text generation, and sentiment analysis. These models, characterized by their large-scale neural net-

work architectures and vast training datasets, have shown remarkable capabilities in understanding and generating human language. The advent of LLMs, such as OpenAI's GPT-4 (Achiam et al. (2023)), and BERT (Devlin et al. (2018)) by Google, has pushed the boundaries of what machines can achieve in linguistic tasks, providing near-human performance in various applications. The parameters and KV cache in Large Language Models are very large in size. For instance, GPT-2 contains 1.5 billion parameters (Solaiman et al. (2019)), whereas GPT-3 has expanded to 175 billion parameters (Brown et al. (2020)). Additionally, the KV cache accounts for over 30% of GPU memory during deployment, compared to the 65% occupied by the parameters (Kwon et al. (2023)). Both the parameters and KV cache are stored in the form of matrices. GPTQ (Frantar et al. (2022)) and SqueezeLLM (Kim et al. (2023)) directly address the QEM problem by treating it as the optimization objective for quantizing parameters and the KV cache. Therefore, the abstracted scientific problem of QEM is highly significant.

There has been a growing literature on the matrix quantization. The first category of methods focuses on independently compressing the elements of a matrix for each specific scenario. Examples of such techniques include LLM.int8 (Ge et al. (2013)), Optimal Brain Damage (LeCun et al. (1989)), GPTQ (Frantar et al. (2022)), AWQ (Lin et al. (2023)), SmoothQuant (Xiao et al. (2023)), and OmniQuant (Shao et al. (2023)). The second category groups matrix elements by columns and then applies quantization to each group. Relevant works in this category include product quantization (PQ) (Jegou et al. (2010)), optimized product quantization (OPQ) (Ge et al. (2013)), locally optimized product quantization (LOPQ) (Kalantidis & Avrithis (2014)), and SqueezeLLM (Kim et al. (2023)). The first category of work can be summarized as utilizing outliers and the importance of elements in specific scenarios to improve the RTN (Round-To-Nearest) [1] (Gray & Neuhoff (1998)) algorithm. The second category focuses on enhancements to PQ algorithms. The primary drawback of the first category is that it does not consider the correlation between elements, instead independently quantizing and storing elements. Conversely, the second category fails to account for the relative order of elements.

In our research, we observed that the order of elements has a significant impact on the quantization outcome. Intuitively, consider a matrix $\begin{bmatrix} 1 & 2 \\ 2 & 1 \end{bmatrix}$. To losslessly quantize this matrix (with MSE=0), two vectors, $[1, 2]$ and $[2, 1]$, are required. However, if the matrix is reordered to $\begin{bmatrix} 1 & 2 \\ 1 & 2 \end{bmatrix}$, it can be losslessly quantized using just a single vector, $[1, 2]$.

Using these ideas, we propose Quantum Entanglement Trees (QET) to address the QEM problem in matrix quantization. The core idea of QET is to leverage the local orderliness of matrix elements to optimize the QEM problem. The design involves swapping adjacent elements of the matrix to form a new, locally ordered matrix. To cover a broader range, we can perform further element swapping on the initially locally ordered matrix, similar to the approach of receptive field used in convolutional neural networks (LeCun et al. (1998)). This step can be repeated for multiple iterations. The newly ordered matrix is then grouped by columns, with each group being quantized. Additionally, we propose two optimization algorithms based on the basic QET. First, the residuals between the original and quantized matrices can be further quantized to enhance the accuracy of the results. Second, the codebook can be quantized to reduce storage requirements.

We evaluate QET on multiple datasets. Experimental results demonstrate that QET can effectively reduce MSE to 5.05%, 13.33%, and 11.89% of the current best method on the LLM dataset, K cache, and V cache, respectively. Our code has been open-sourced on GitHub ( QET (2024)).

We summarize our contributions below.

**Abstracted a problem**: We abstracted the Quantization Error Minimization (QEM) problems from various application scenarios.

**Designed an algorithm**: We developed the Quantum Entanglement Trees (QET) algorithm, leveraging the concept of local orderliness to optimize the QEM problem.

---

[1]RTN (Round-To-Nearest) is an algorithm that quantizes elements by rounding each value to its nearest representable level.

**Proposed two optimizations**: We introduced residual quantization to reduce MSE and codebook quantization techniques to optimize memory efficiency.

## 2 PROBLEM SETTING

In this section, we formally define the Quantization Error Minimization (QEM) problem and analyze the impact and feasibility of rearranging matrix elements prior to quantization.

**QEM Problem:** we denote the matrix to be quantized as $X$, the quantized matrix as $X^q$, and the matrix obtained by dequantizing $X^q$ as $X'$. Both $X$ and $X'$ are $n \times d$ dimensional, but the dimensions of $X^q$ depend on the quantization algorithm. The elements of these matrices are denoted as $x_{(i,j)}$, $x^q_{(i,j)}$, and $x'_{(i,j)}$, respectively. The memory occupied by a matrix is denoted as memory(), and the memory constraint as mem_constrain.

**Definition 1** (Quantization Error Minimization (QEM) Problem)**.** The Quantization Error Minimization (QEM) problem is defined as the task of minimizing the combined objective of the Mean Squared Error (MSE) and the memory size of the quantized matrix:

$$\text{minimize MSE}(X, X') + \lambda \cdot \text{memory}(X^q),$$

where

$$\text{MSE}(X, X') = \frac{1}{n \cdot d} \sum_{i,j} \left( x_{(i,j)} - x'_{(i,j)} \right)^2,$$

$\lambda$ is a regularization parameter that balances the trade-off between minimizing the MSE and the memory usage of the quantized matrix.

Subject to the condition that the quantized matrix occupies the same space:

$$\text{memory}(X^q) \leq \text{mem\_constrain},$$

**Rearranging matrix elements:** Our idea is to rearrange the matrix elements before quantization. Let the quantization algorithm be denoted as quant() and the rearrangement algorithm as rearrange(). Previously, the quantized matrix was obtained as $X^q = \text{quant}(X)$, whereas in our rearrangement approach, it is obtained as $X^q = \text{quant}(\text{rearrange}(X))$. Therefore, the rearrangement algorithm only affects the quantization if the quantization method is order-sensitive.For instance, order-insensitive algorithms like RTN do not benefit from rearrangement because RTN processes each element independently, while order-sensitive algorithms like PQ can be optimized through rearrangement.

However, finding the optimal arrangement requires exploring a large search space. Specifically, for an $n \times d$ matrix, there are $(n!)^d$ possible rearrangements. Therefore, searching the entire space is impractical, and we need heuristic algorithms to prioritize exploring more promising regions of the search space.

## 3 METHODS

The core idea of Quantum Entanglement Trees (QET) is to leverage the local orderliness of matrix elements by rearranging adjacent elements to optimize the matrix quantization algorithm. Additionally, multiple iterations are used to expand the algorithm's swapping field. We begin by describing the basic version of this idea and then propose two optimizations for the basic algorithm.

### 3.1 BASIC ALGORITHM

The steps and design of the QET algorithm are illustrated in Figure 1. The figure provides a conceptual overview of the Quantum Entanglement Trees (QET) algorithm.

On the left side of the figure, the process of rearranging matrix elements through multiple iterations in the QET is depicted. This iterative procedure involves comparing adjacent elements and ensuring that adjacent elements are not placed in the same matrix. This separation of adjacent elements into different matrices is conceptually similar to quantum entanglement, hence the name "Quantum Entanglement Trees." The QET algorithm performs multiple iterations, with each iteration comparing

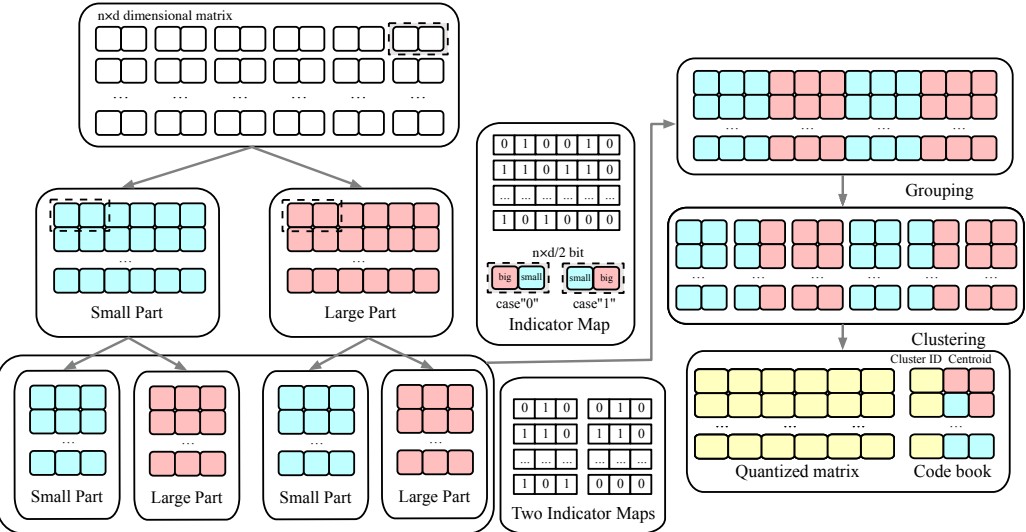

Figure 1: QET algorithm.

adjacent elements and splitting them based on their values. This approach is based on two observations. First, the local orderliness of the matrix can enhance the regularity of matrix elements, thereby improving compression efficiency. Second, multiple iterations allow the orderliness to cover a larger number of elements. For instance, adjacent elements after the first iteration might be non-adjacent in the original matrix. This iterative process, similar to that of convolutional neural networks, extends the "receptive field" to cover more elements.

On the right side of the figure, the QET compression process is shown. After the elements have been rearranged, the algorithm groups the sub-matrices, followed by clustering to determine the centroids. These centroids are used to quantize the matrix. The final quantized matrix, codebook, and indicator maps are the outputs of the QET algorithm. This divide-and-conquer compression method is based on an observation: by splitting the matrix, the expressive power can be increased under a fixed storage constraint. For example, if the matrix is divided into $m$ blocks, each with $c$ centroids, the entire matrix can express $c^m$ centroids while only storing $c \times m$ centroids. Without splitting, storing $c^m$ centroids would be required.

## 3.2 QET QUANTIZATION

The proposed QET Algorithm is designed to optimize the quantization of a matrix $X \in \mathbb{R}^{n \times d}$ by leveraging local orderliness. The algorithm proceeds in four key steps, as shown in Algorithm 2:

**Step 1: Initial Partitioning.** The algorithm starts by comparing each adjacent pair of elements $(x_{i,j}, x_{i,j+1})$ in the input matrix $X$. Based on the comparison of these elements, the smaller value is placed into matrix $S$ and the larger into matrix $L$. Concurrently, an Indicator Map $I$ is generated, where a 0 denotes that the smaller element is on the left and a 1 denotes it is on the right. This process results in the creation of two matrices, $S$ and $L$, which collectively represent the initial partitioning of the original matrix $X$.

**Step 2: Recursive Partitioning.** Following the initial partitioning, the algorithm undertakes a recursive partitioning process. Starting from $k = 1$, the matrices $S_k^i$ and $L_k^i$ [2] derived from the previous iteration (for $i = 1, 2, \ldots, 2^{k-1}$) are further partitioned based on their sizes. In each iteration, a new Indicator Map $I_{k+1}$ is generated and then merged with the existing Indicator Maps. This recursive process continues until the predefined number of iterations $l$ is reached. The resulting Indicator Maps $I_1, I_2, \ldots, I_l$ are stored in the set $IM$. The final matrices $S_l$ and $L_l$ are combined to form a new matrix $X^*$, which represents a locally ordered version of the original matrix.

---

[2]Since each level has multiple $S$ and $L$ matrices, we use the subscript to denote the iteration number and the superscript to denote the sequence within each iteration.

**Step 3: Subspace Grouping.** After constructing the matrix $X^*$, it is divided into subspaces $\{G_i\}$, where $i = 1, 2, \ldots, m$. This division is based on the local ordering of elements, which facilitates more effective clustering and quantization in the subsequent steps.

**Step 4: Clustering and Quantization.** In the final step, each subspace $G_k$ (for $k = 1, 2, \ldots, m$) is clustered to determine a set of centroids $C_k = \{\mathbf{c}_j\}$. Each vector $\mathbf{g_i}$ within the subspace is then quantized by assigning it to the nearest centroid. This process results in the quantized matrix $X^q$. The output of the algorithm includes the quantized matrix $X^q$, the codebook $\mathbf{C}$ containing the centroids, and the Indicator Maps $IM$.

QET Dequantization is the inverse process of QET Quantization. We provide the detailed description in Appendix A.1.

---

**Algorithm 1** QET Quantization Algorithm

---

1: **Input:** Matrix $X \in \mathbb{R}^{n \times d}$, $m$ is the number of subspaces for quantization, $l$ is the number of QET iterations.
2: **Output:** Quantized matrix $X^q = \{X_{i,k}^q\}$ where $i = 1, \ldots, d$ and $k = 1, \ldots, m$, Codebook $\mathbf{C} = \{C_i\}$ where $i = 1, \ldots, m$, and Indicator Maps $IM = \{I_i\}$ where $i = 1, \ldots, l$
3: Initialize $IM$ and $X^q$ as empty
4: **Step 1: Initial Partitioning**
5: **for** each adjacent pair $(x_{i,j}, x_{i,j+1})$ in $X$ **do**
6:     **if** $x_{i,j} > x_{i,j+1}$ **then**
7:         $S_{i,j} \leftarrow x_{i,j+1}, L_{i,j} \leftarrow x_{i,j}$
8:         $I\left(\frac{i}{2}, j\right) \leftarrow 0$
9:     **else**
10:         $S_{i,j} \leftarrow x_{i,j}, L_{i,j} \leftarrow x_{i,j+1}$
11:         $I\left(\frac{i}{2}, j\right) \leftarrow 1$
12:     **end if**
13: **end for**
14: **Step 2: Recursive Partitioning**
15: $k \leftarrow 1$
16: **while** $k \neq l$ **do**
17:     **for** $i = 1$ to $2^{k-1}$ **do**
18:         Apply size-based partitioning to $S_k^i$ and $L_k^i$
19:         Generate and merge new Indicator Maps into $I_{k+1}$ for each partition
20:     **end for**
21:     $k \leftarrow k + 1$
22: **end while**
23: Store $I_1, I_2, \ldots, I_l$ into $IM$
24: $X^* \leftarrow S_l^1 \cup L_l^1 \cup S_l^2 \cup L_l^2 \cup \cdots \cup S_l^{2^{l-1}} \cup L_l^{2^{l-1}}$ ▷ Combine the locally ordered parts from the final layer $l$
25: **Step 3: Subspace Grouping**
26: Group the matrix $X^*$ into subspaces $\{G_i\}$ where $i = 1, 2, \ldots, m$
27: **Step 4: Clustering and Quantization**
28: Apply clustering to find centroids $C_k = \{\mathbf{c}_j\}$ for $G_k$
29: **for** each vector $\mathbf{g_i}$ in $G_k$ (for $k = 1, 2, \ldots, m$) **do**
30:     $X_{i,k}^q \leftarrow \arg\min_{\mathbf{c}_j} \|\mathbf{g_i} - \mathbf{c}_j\|$         ▷ Quantize to nearest centroid within group
31: **end for**
32: **Return** Quantized matrix $X^q$, Codebook $\mathbf{C}$, and Indicator Maps $IM$

---

## 3.3 Theoretical Guarantees of QET

In this section, we will first prove that the MSE of our QET algorithm is lower than that of PQ. Then, we will compare the time complexity of the algorithms.

**Theorem 1.** For the QET algorithm without optimization, where matrix elements are independently sampled from a normal distribution $x_{(i,j)} \sim \mathcal{N}(\mu, \sigma^2)$, the Mean Squared Error (MSE) is:

$$\text{MSE}_{\text{QET}} = 0.682 \times \text{MSE}_{\text{PQ}}.$$

The detailed proof can be found in Appendix A.2.

**Theorem 2. (Time Complexity for Quantization)** The QT of QET is reduced compared to PQ by a factor of approximately $2^{\Delta}$. Specifically, the ratio of QT is

$$\frac{QT_{\text{QET}}}{QT_{\text{PQ}}} \approx 2^{-\Delta},$$

where $QT_{\text{QET}}$ represents the quantization time for QET, while $QT_{\text{PQ}}$ represents the quantization time for PQ.

The detailed proof can be found in Appendix A.3.

Theorem 1 shows that when the matrix elements follow a normal distribution, the MSE is reduced to 0.682 times its original value. For any distribution, as long as the variance of the matrix elements is reduced, the MSE will also decrease (Theorem 1 in Appendix A.2). Theorem 1 demonstrates that our algorithm has a shorter quantization time. Furthermore, according to Theorem 3 in Appendix A.4, the dequantization time increases.

### 3.4 OPTIMIZATIONS

In this section, we propose three optimizations: Residual Quantization Optimization (RQO) and Codebook Quantization Optimization (CQO).

#### 3.4.1 RESIDUAL QUANTIZATION OPTIMIZATION

Residual Quantization Optimization (RQO) is grounded in our observation that the data range of the quantized matrix is significantly reduced after applying the QET algorithm. As shown in Table 1, our experiments indicate that at a compression ratio[3] of 12, the data range of the quantized matrix is reduced to 9.2% of the original matrix's range.

Table 1: Range Ratios, and Codebook(CB) Proportions in different compression ratios.

| Compression Ratio | 4.0 | 8.0 | 12.0 |
|---|---|---|---|
| Range Ratio (%) | 1.6 | 4.0 | 9.2 |
| CB Proportion (%) | 87.5 | 78.1 | 71.8 |

Moreover, with a compression ratio of 4, the data range of the quantized matrix is reduced to approximately 1.6% of the original matrix's range.

Matrix quantization algorithms are more efficient at compressing matrix with a smaller data range; therefore, we propose Residual Quantization Optimization (RQO). The RQO begins with the quantization of the original matrix to produce a quantized matrix. This matrix is then dequantized to yield a dequantized matrix, which is subtracted from the original matrix to form a residual matrix. The residual matrix typically exhibits a significantly reduced data range, making it easier to compress. This residual matrix is then quantized using a chosen matrix quantization algorithm, such as RTN or QET.

During the dequantization process, the quantized residual is dequantized, and this dequantized residual is added back to the initially dequantized matrix to produce a revised dequantized matrix.

#### 3.4.2 CODEBOOK QUANTIZATION OPTIMIZATION

The codebook occupies a significant portion of the space in the QET algorithm. As shown in Table 1, our experiments indicate that when the compression ratio is 12, the codebook occupies 71.8% of the total space, and when the compression ratio is 4, the codebook occupies 87.5% of the total space. Therefore, compressing the codebook can significantly reduce the space used by the algorithm. We proposed Codebook Quantization Optimization (CQO). Our optimization mainly employs the RTN method to compress the codebook.

## 4 EXPERIMENTS

### 4.1 EXPERIMENT SETUP

**Dataset:** To evaluate the effectiveness of our proposed algorithm, we conducted experiments using three synthetic datasets and two real-world datasets: the synthetic normal distribution datasets, LLM

---

[3]Compression ratio = $\frac{\text{Space occupied by the matrix before quantization}}{\text{Space occupied by the matrix after quantization}}$

weight dataset and KV cache dataset. Below, we provide a detailed description of each dataset. Each element in the matrix is represented with 32 bits, except for the KV cache dataset, where each element is represented with 16 bits.

(1) Synthetic normal distribution dataset: This dataset was generated by drawing each element of the matrix from a truncated normal distribution with a mean of 0.5 and a standard deviation of 0.16. Additionally, one out of every ten thousand elements was replaced with an outlier value, randomly chosen between -100 and 100. For the synthetic normal distribution dataset, we generated three types of matrices of different sizes: synthetic dataset 1: $1024 \times 128$ matrices, synthetic dataset 2: $1024 \times 512$ matrices, and synthetic dataset 3: $1024 \times 1024$ matrices.

(2) LLM weight dataset: This dataset comprises weight matrices extracted from the large language model (LLM) LLaMA2 (Touvron et al. (2023)). The sizes of the LLM weight matrices are $11008 \times 4096$.

(3) KV cache dataset: The KV cache dataset is derived from the key-value pairs stored in the cache during inference in KV Quant (Hooper et al. (2024)). The key and value are stored in matrices, respectively. In the KV cache dataset, the sizes of both the K matrices and the V matrices are $4096 \times 4096$.

**Platform and implementation:** We conducted our algorithm evaluations on a high-performance server equipped with an Intel Xeon Platinum 6462C (Sapphire Rapids) processor, featuring 16 virtual CPUs (vCPUs), operating at a base frequency of 3.3 GHz, with a maximum turbo frequency of 3.9 GHz. The server also includes 64GB of memory, providing robust computational capabilities. All algorithms were implemented and executed on this server environment to ensure optimal performance for our experimental evaluations.

**Metrics:** We primarily measure the accuracy and time consumption of the algorithm. We use MAE (Mean Absolute Error), MRE (Mean Relative Error), and MSE (Mean Squared Error) as accuracy metrics. Below, we introduce these metrics in detail.

Let $x_{(i,j)}$ denote the elements of the original matrix to be quantized, and $x'_{(i,j)}$ denote the elements of the dequantized matrix.

$$\text{MAE} = \frac{1}{n \cdot d} \sum_{i,j} \left| x_{(i,j)} - x'_{(i,j)} \right|, \ \text{MSE} = \frac{1}{n \cdot d} \sum_{i,j} \left( x_{(i,j)} - x'_{(i,j)} \right)^2$$

For time efficiency, we use Quantization Time (QT) and DeQuantization Time (DQT) as the metrics.

**Comparative Algorithms:** For the abstract Quantization Error Minimization (QEM) problem, our comparative algorithms fall into two main categories. The first category involves independently compressing the elements of a matrix for each specific scenario, which can be abstracted as the RTN (Round-To-Nearest) algorithm. The second category groups matrix elements by columns and then applies quantization to each group. Related algorithms in this category include PQ, OPQ, and LOPQ. Therefore, our comparative algorithms are as follows: RTN (Round-To-Nearest) (Gray & Neuhoff (1998)), PQ (Product Quantization) (Jegou et al. (2010)), OPQ (Optimized Product Quantization) (Ge et al. (2013)), and LOPQ (Locally Optimized Product Quantization) (Kalantidis & Avrithis (2014)).

**Parameter Selection:** We first introduce the parameter settings of the QET algorithm. For an $n \times d$ matrix with $a$ bit per element, we perform a quantization operation with a compression ratio of $\theta$. The QET algorithm iterates $l$ rounds. For residual optimization, we perform $r$ residual iterations. During the $i$th residual step, the number of clusters for centroids is $k_i$, and the number of groups during the grouping operation is $m$. For codebook quantization optimization, the bit length of the codebook after compression is $a'$ bit. $R_i$ denotes the ratio of the space occupied by the codebook and the quantized matrix to the total space excluding the indicator maps during the $i$th iteration of residual optimization. To ensure that the total memory usage remains within the available memory limits, we have the following constraints on the parameters of the QET algorithm.

$$k_i \times d \times a' + 2a + n \times m \times \log_2(k_i) \leq \left[ \frac{n \times d \times a}{\theta} - 0.5 \times n \times d \times l \right] \times R_i, \quad \text{for } i = 1, \ldots, r.$$

Given the parameters $d$, $a'$, $a$, $n$, $m$, $\theta$, $l$, and $R_i$, we calculate the value of $k_i$ using the bisection method. Unless otherwise specified, the QET parameters are configured as follows: $n$, $d$, and $a$

are determined by the input matrix. We perform a standard QET and one residual QET, both with codebook quantization. $r = 2$, the first QET has 3 iterations, and the second QET does not perform matrix permutation ($l = 3$). Additionally, $d/m = 8$, $a' = 10$, $R_1 = 70\%$, and $R_2 = 30\%$.

For RTN, PQ, OPQ, and LOPQ, under the same space constraints, we use the same common parameters as our algorithm, while other parameters are set according to the recommended configurations from their respective papers. We define the names of different versions of the algorithm as follows: **Vanilla** refers to the basic QET algorithm without any optimizations. **Vanilla_CQO** refers to the Vanilla algorithm with codebook quantization optimization, where the codebook is quantized using the RTN algorithm. **Vanilla_RQO** indicates the Vanilla algorithm with residual optimization, where the residuals are processed using the QET algorithm. Finally, **QET** is the Vanilla algorithm with both residual optimization and codebook quantization optimization (including all optimizations), where the codebook is quantized using the RTN algorithm and the residuals are processed using the QET algorithm.

### 4.2 MATRIX QUANTIZATION RESULT

In this section, we provide a comprehensive comparison of the accuracy and computational efficiency of the proposed QET algorithm against other benchmark methods. We will first discuss the precision metrics (MAE and MSE) across different datasets and compression ratios, followed by an analysis of the time metrics (QT and DQT) to evaluate the efficiency of each algorithm. Some algorithms are unable to run on certain datasets or at specific compression ratios. For instance, in the case of square matrices, the rotation matrices for OPQ and LOPQ become as large as the original matrix, making compression impossible. In such cases, we observe missing data points or curves in the corresponding figures where the algorithms could not produce results.

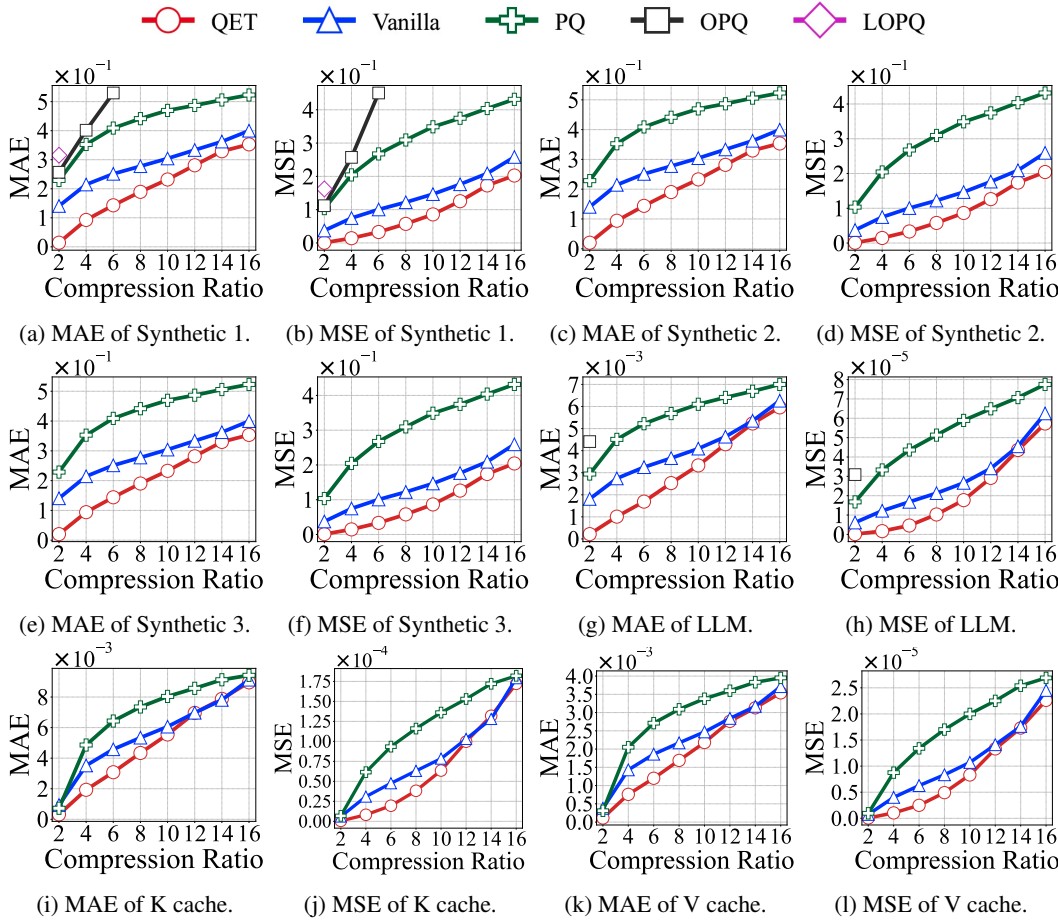

Figure 2: MAE and MSE of different datasets.

**MAE and MSE:** In our experiments, we evaluated the performance of our quantization algorithm across multiple datasets using MAE and MSE. These metrics provide a comprehensive understanding of the algorithm's accuracy at different compression ratios, ranging from 2 to 16. The average MSE of RTN for the synthetic dataset 1, synthetic dataset 2, synthetic dataset 3, LLM dataset, K cache dataset, and V cache dataset is 71.74 times, 334.59 times, 247.39 times, 12.56 times, 10.57 times, and 7.69 times that of QET, respectively. Therefore, we do not include RTN in the figures for comparison.

For synthetic dataset 1: As shown in Figures 2a and 2b , our QET algorithm consistently outperforms the baseline algorithms across all compression ratios. The MAE and MSE values increase as the compression ratio rises, but QET maintains a significantly lower error compared to PQ and OPQ algorithms. The Vanilla version, a faster variant of QET, shows slightly higher error than the full QET version, but still remains competitive. For example, at a compression ratio of 4, compared to the best-performing algorithm, PQ, QET reduces the MSE to 6.94%, while the Vanilla version reduces the MSE to 36.53%.

For Synthetic Dataset 2: In Figures 2c and 2d, a similar trend is observed with Synthetic Dataset 2. The QET algorithm achieves the lowest MAE and MSE values, demonstrating its robustness across different types of synthetic data. Notably, the gap between QET and the other methods becomes more pronounced as the compression ratio increases, indicating the superiority of QET in handling higher compression rates without significantly sacrificing accuracy. Due to the space consumption of OPQ and LOPQ, they were unable to handle the compression of square matrices in this dataset. The vanilla version performs similarly to the QET algorithm. For example, at a compression ratio of 4, compared to the best-performing algorithm, PQ, QET reduces the MSE to 7.13%, while the Vanilla version reduces the MSE to 36.51%.

For Synthetic Dataset 3: Figures 2e and 2f show the results for the third synthetic dataset. While the errors increase with higher compression ratios, QET still outperforms the other algorithms. The difference in performance becomes especially significant at higher compression ratios (e.g., 12 to 16), where QET and vanilla continue to show lower error rates. For example, at a compression ratio of 4, compared to the best-performing algorithm, PQ, QET reduces the MSE to 7.21%, while the Vanilla version reduces the MSE to 36.54%.

For the LLM dataset, as depicted in Figures 2g and 2h, the QET algorithm demonstrates superior performance. The MAE and MSE values for QET are consistently lower than those for Vanilla, PQ, and OPQ, indicating that QET is well-suited for compressing large-scale language model weights without significantly sacrificing accuracy. At higher compression ratios (13 to 16), the MAE and MSE of Vanilla are similar to those of QET. For example, at a compression ratio of 4, compared to the best-performing algorithm, PQ, QET reduces the MSE to 5.05%, while the Vanilla version reduces the MSE to 36.64%.

For KV Cache Dataset: The results of K cache is shown in Figures 2i and 2j. The results of V cache is shown in Figures 2k and 2l. Similar to the LLM datasets, the QET algorithm achieves the best results, with lower MAE and MSE values across all compression ratios. At compression ratios between 13 and 16, the MAE and MSE values of Vanilla closely approach those of QET. For example, at a compression ratio of 4 for K cache, compared to the best-performing algorithm, PQ, QET reduces the MSE to 13.33%, while the Vanilla version reduces the MSE to 50.24%. And at a compression ratio of 4 for V cache, compared to the best-performing algorithm, PQ, QET reduces the MSE to 11.89%, while the Vanilla version reduces the MSE to 45.83%.

In summary, the QET algorithm consistently outperforms baseline methods, achieving lower MAE and MSE values across all datasets and compression ratios. Vanilla also demonstrates relatively low MAE and MSE values, and on certain datasets, it performs comparably to QET at higher compression ratios.

**QT and DQT:** In this experiment, we evaluated the performance of the quantization process by measuring both Quantization Time (QT) and Dequantization Time (DQT) across multiple datasets.

For QT, we observed that the QET algorithm achieves efficient quantization times at higher compression ratios. The Vanilla method consistently achieves the lowest QT across all datasets, making it the faster option, albeit with a slight loss in accuracy. Regarding DQT, QET maintains stable performance across all datasets. The Vanilla method achieves faster dequantization but is still slower

than PQ. Meanwhile, OPQ and LOPQ exhibit slower dequantization speeds. For a detailed analysis, please refer to Appendix A.5.

### 4.3 Ablation Studies and Discussions

These experiments demonstrate the effectiveness of the algorithm optimizations. Specifically, both Residual Quantization Optimization (RQO) and Codebook Quantization Optimization (CQO) improve the baseline Vanilla algorithm and provide comparisons with the QET algorithm. The ablation studies primarily focus on accuracy improvements. We evaluated the performance on the LLM dataset by measuring both MAE and MSE.

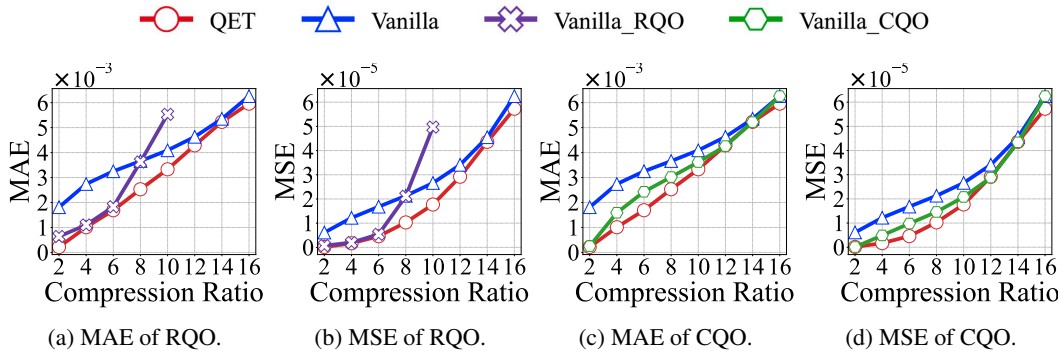

(a) MAE of RQO.  (b) MSE of RQO.  (c) MAE of CQO.  (d) MSE of CQO.

Figure 3: Ablation Studies of RQO and CQO.

**Residual Quantization Optimization (RQO):** As illustrated in Figures 3a and 3b, the introduction of RQO into the Vanilla algorithm leads to a notable reduction in MAE and MSE values. At lower compression ratios, Vanilla with RQO achieves performance that is very close to QET, particularly in terms of MSE. The gap widens at higher compression ratios because, while the storage space remains the same for both algorithms, RQO requires more codebooks due to the multi-layer structure, which reduces the number of centroids in clustering. On the other hand, QET optimizes codebook compression, allowing for more centroids and better MAE and MSE performance at higher compression ratios. For instance, at a compression ratio of 4, the RQO optimization reduces MSE to 16.23% of the original value.

**Codebook Quantization Optimization (CQO):** In Figures 3c and 3d, the impact of CQO on the Vanilla algorithm is shown. CQO helps to decrease MAE and MSE, especially at lower compression ratios. CQO optimization effectively reduces the size of the codebook, leading to improvements in both MAE and MSE. However, at higher compression ratios, the accuracy improvements become less noticeable because the codebook size is already small at these higher compression levels, leaving limited room for further compression. For instance, at a compression ratio of 4, the CQO optimization reduces MSE to 40.08% of the original value.

## 5 Conclusion

Matrix quantization involves representing matrix elements in a more space-efficient form to reduce storage usage, with dequantization restoring the original matrix for practical applications. We frame the Quantization Error Minimization (QEM) problem as minimizing the distance between a matrix before and after quantization, constrained by the condition that the quantized matrix occupies the same memory space.

To address the QEM problem, we propose Quantum Entanglement Trees (QET), which leverage the local orderliness of matrix elements through iterative element swapping to form a locally ordered matrix. To enhance the QET algorithm, we introduce two key optimizations: residual quantization optimization and codebook quantization optimization.

Our experimental results demonstrate that QET can effectively reduce MSE to 5.05%, 13.33%, and 11.89% of the best existing methods on the LLM dataset, K cache, and V cache, respectively.

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

# A  Appendix

## A.1  QET Dequantization

The dequantization process, as described by the QET Dequantization Algorithm, aims to reconstruct the original matrix $X'$ from its quantized version $X^q$. This process involves several key steps, as shown in Algorithm 2:

**Step 1: Dequantization Using Codebook.** The first step involves using the codebook $\mathbf{C}$ to directly retrieve the original vectors corresponding to each quantized vector $\mathbf{g_i^q}$ in the matrix $X^q$. For each vector $\mathbf{g_i^q}$ in $X_{i,k}^q$, the algorithm assigns $\mathbf{g_i'}$ to the centroid $\mathbf{c}_j$ from the codebook, where $j$ is the index corresponding to the quantized vector. This step effectively reverses the quantization by mapping the quantized vectors back to their original forms using the codebook entries.

**Step 2: Reverse Subspace Grouping.** After mapping, the algorithm combines the dequantized subspaces $\{G_i'\}$, where $i = 1, 2, \ldots, m$, into a single matrix $X'^*$. Each subspace $G_i'$ contains the vectors $\mathbf{g_i'}$ corresponding to the original matrix entries. This step reconstructs the subspaces into an uniform matrix that is a preliminary version of the original matrix $X'$.

**Step 3: Reverse Recursive Partitioning and Final Reconstruction.** The final step involves reversing the recursive partitioning process that was applied during quantization. Starting from the last iteration $k = l$, the algorithm merges the matrices $S_k$ and $L_k$ based on the Indicator Maps $I_k$ and the matrix $X'^*$. This merging process is iteratively applied from $k = l$ down to $k = 1$, updating $X'^*$ at each step to gradually reconstruct the original matrix structure. The final output of the algorithm is the reconstructed matrix $X' \in \mathbb{R}^{n \times d}$, which approximates the original matrix before quantization.

---

**Algorithm 2** QET Dequantization Algorithm

---

1: **Input:** Quantized matrix $X^q = \{X_{i,k}^q\}$, where $i = 1, \ldots, d$ and $k = 1, \ldots, m$; Codebook $\mathbf{C} = \{C_i\}$, where $i = 1, \ldots, m$; and Indicator Maps $IM = \{I_i\}$, where $i = 1, \ldots, l$
2: **Output:** Reconstructed matrix $X' \in \mathbb{R}^{n \times d}$
3: Initialize $X'$ as empty
4: **Step 1: Dequantization Using Codebook**
5: **for** each vector $\mathbf{g_i^q}$ in $X_{i,k}^q$ **do**
6:     $\mathbf{g_i'} \leftarrow \mathbf{c}_j$, where $j$ corresponds to the codebook entry for $\mathbf{g_i^q}$ ▷ Direct lookup from codebook
7: **end for**
8: **Step 2: Reverse Subspace Grouping**
9: Combine subspaces $\{G_i'\}$, where $i = 1, 2, \ldots, m$, $G_i' = \{\mathbf{g_i'}\}$ for $i = 1$ to $n$, to form $X'^*$
10: **Step 3: Reverse Recursive Partitioning**
11: $k \leftarrow l$
12: **while** $k \neq 0$ **do**
13:     **for** $i = 1$ to $2^{k-1}$ **do**
14:         Merge $S_k^i$ and $L_k^i$ based on $I_k$ and $X'^*$, then update the results into $X'^*$
15:     **end for**
16:     $k \leftarrow k - 1$
17: **end while**
18: **Return** Reconstructed matrix $X'$

---

## A.2  Proof of Theorem 1 (Theoretical analysis of MSE)

**Lemma 1.** Suppose each element in the matrix is independently sampled with a mean $\mu$ and variance $\sigma^2$. After quantization by the PQ algorithm, the Mean Squared Error (MSE) is given by:

$$\text{MSE}_{\text{PQ}} = \sigma^2 \left( 1 - \frac{1}{n_k} \right).$$

*Proof.* Let $c_{(i,j)}$ denote the cluster centroid matrix, and let $n_k$ be the number of vectors in the cluster. Define the residual matrix as $s_{(i,j)}$.

The expectation of the residual matrix is:
$$E[s_{(i,j)}] = E[x_{(i,j)} - c_{(i,j)}] = \mu - \mu = 0.$$

The Mean Squared Error (MSE) can be expressed as the variance of the residual matrix:
$$\text{MSE}_{\text{PQ}} = \text{Var}[s_{(i,j)}] = \text{Var}[x_{(i,j)}] + \text{Var}[c_{(i,j)}] - 2 \cdot \text{Cov}(x_{(i,j)}, c_{(i,j)}).$$

Given that $c_{(i,j)}$ is the centroid within the cluster:
$$\text{MSE}_{\text{PQ}} = \sigma^2 + \frac{\sigma^2}{n_k} - 2E\left[(x_{(i,j)} - \mu)\left(\frac{1}{n_k}\sum_{i=1}^{n_k} x_{(i,j)} - \mu\right)\right]$$

$$= \sigma^2 + \frac{\sigma^2}{n_k} - 2 \cdot \frac{1}{n_k}\left(E\left[(x_{(i,j)} - \mu)^2\right] + \sum_{i \neq i'} E\left[(x_{(i,j)} - \mu)(x_{i'j} - \mu)\right]\right).$$

Since $x_{(i,j)}$ and $x_{i'j}$ (for $i \neq i'$) are independent:
$$\text{MSE}_{\text{PQ}} = \sigma^2 + \frac{\sigma^2}{n_k} - 2 \cdot \frac{\sigma^2}{n_k} = \left(1 - \frac{1}{n_k}\right)\sigma^2. \tag{1}$$

$\square$

**Theorem 1. (Theoretical analysis of MSE)** For the QET algorithm without optimization, where matrix elements are independently sampled from a normal distribution $x_{(i,j)} \sim \mathcal{N}(\mu, \sigma^2)$, the Mean Squared Error (MSE) is:
$$\text{MSE}_{\text{QET}} = 0.682 \times \text{MSE}_{\text{PQ}}.$$

*Proof.* We denote by $F_Y(y)$ the cumulative distribution function (CDF) of the elements on the right side of the matrix (i.e., the larger elements).

$$F_Y(y) = P(Y \leq y) = P(\max(X_l, X_r) \leq y) = P(X_l \leq y) \cdot P(X_r \leq y)$$

Where $y$ represents the elements of the matrix formed by the larger of the adjacent elements in matrix $X$. $X_l$ and $X_r$ denote the random variables corresponding to the adjacent left and right elements in matrix $X$, respectively. Since $X_l$ and $X_r$ follow a normal distribution $\mathcal{N}(\mu, \sigma^2)$, we have:

$$F_Y(y) = \Phi\left(\frac{y-\mu}{\sigma}\right) \cdot \Phi\left(\frac{y-\mu}{\sigma}\right) = \left[\Phi\left(\frac{y-\mu}{\sigma}\right)\right]^2 = \frac{1}{4}\left[1 + \text{erf}\left(\frac{y-\mu}{\sigma\sqrt{2}}\right)\right]^2$$

Take the derivative of $F_Y(y)$ with respect to $y$:

$$f_Y(y) = \left[1 + \text{erf}\left(\frac{y-\mu}{\sigma\sqrt{2}}\right)\right] \cdot \frac{1}{\sqrt{2\pi\sigma^2}}\exp\left(-\frac{(y-\mu)^2}{2\sigma^2}\right)$$

The mean and variance of $Y$ are given by:

$$E(Y) = \int_{-\infty}^{\infty} y\left[1 + \text{erf}\left(\frac{y-\mu}{\sigma\sqrt{2}}\right)\right] \cdot \frac{1}{\sqrt{2\pi\sigma^2}}\exp\left(-\frac{(y-\mu)^2}{2\sigma^2}\right) dy$$

$$Var(Y) = \int_{-\infty}^{\infty} (y - E(Y))^2\left[1 + \text{erf}\left(\frac{y-\mu}{\sigma\sqrt{2}}\right)\right] \cdot \frac{1}{\sqrt{2\pi\sigma^2}}\exp\left(-\frac{(y-\mu)^2}{2\sigma^2}\right) dy$$

Using numerical integration methods, we obtain:
$$\mathbb{E}(Y) = \mu + 0.564\sigma, \quad \text{Var}(Y) = 0.682\sigma^2.$$

Because QET can be considered a special case of the PQ algorithm, according to Equation 1,

$$\text{MSE}_{\text{QET}} = \left(1 - \frac{1}{n_k}\right)\text{Var}(Y) = 0.682\left(1 - \frac{1}{n_k}\right)\sigma^2 = 0.682 \cdot \text{MSE}_{\text{PQ}}. \tag{2}$$

Taking symmetry into account, this conclusion also holds for the smaller matrix on the left. $\square$

A.3 THE PROOF OF THEOREM 2 (TIME COMPLEXITY FOR QUANTIZATION)

**Lemma 2.** Let $k_{\text{QET}}$ be the number of centroids in QET and $k_{\text{PQ}}$ be the number of centroids in the Product Quantization (PQ) algorithm, under the same compression ratio. Due to the additional storage required for $l$ indicator matrices in our method, the relationship between $k_{\text{QET}}$ and $k_{\text{PQ}}$ is given by

$$k_{\text{QET}} = \frac{k_{\text{PQ}}}{2^{\Delta}},$$

where

$$\Delta = \frac{l \times (d-1)}{m}.$$

*Proof.* Under the same compression ratio $b$, the total storage requirements for both methods are equal.

For PQ:

$$Memory_{\text{PQ}} = n \times m \times \log_2(k_{\text{PQ}}) + k_{\text{PQ}} \times d \times a.$$

For QET:

$$Memory_{\text{QET}} = n \times m \times \log_2(k_{\text{QET}}) + k_{\text{our}} \times d \times a + l \times n \times (d-1).$$

Setting $Memory_{\text{PQ}} = Memory_{\text{QET}}$ and simplifying:

$$n \times m \times [\log_2(k_{\text{QET}}) - \log_2(k_{\text{PQ}})] + (k_{\text{QET}} - k_{\text{PQ}}) \times d \times a + l \times n \times (d-1) = 0.$$

Assuming $n$ is large and $k_{\text{QET}}, k_{\text{PQ}} \ll n$, the second term is negligible:

$$n \times m \times [\log_2(k_{\text{QET}}) - \log_2(k_{\text{PQ}})] + l \times n \times (d-1) \approx 0.$$

Divide both sides by $n$:

$$m \times [\log_2(k_{\text{QET}}) - \log_2(k_{\text{PQ}})] + l \times (d-1) = 0.$$

Solving for $\log_2(k_{\text{QET}})$:

$$\log_2(k_{\text{QET}}) = \log_2(k_{\text{PQ}}) - \frac{l \times (d-1)}{m}.$$

Therefore,

$$k_{\text{QET}} = \frac{k_{\text{PQ}}}{2^{\Delta}}, \quad \text{where} \quad \Delta = \frac{l \times (d-1)}{m}.$$

$\square$

**Theorem 2. (Time Complexity for Quantization)** The QT of QET is reduced compared to PQ by a factor of approximately $2^{\Delta}$. Specifically, the ratio of QT is

$$\frac{QT_{\text{QET}}}{QT_{\text{PQ}}} \approx 2^{-\Delta},$$

where $QT_{\text{QET}}$ represents the quantization time for QET, while $QT_{\text{PQ}}$ represents the quantization time for PQ.

*Proof.* The QT primarily depends on the clustering step.

QET's Clustering Time:

$$T_{\text{cluster\_QET}} = O\left(t \times n \times k_{\text{QET}} \times d\right).$$

PQ's Clustering Time:

$$T_{\text{cluster\_PQ}} = O\left(t \times n \times k_{\text{PQ}} \times d\right).$$

According to Lemma 2:

$$k_{\text{QET}} = \frac{k_{\text{PQ}}}{2^{\Delta}} \implies \frac{k_{\text{QET}}}{k_{\text{PQ}}} = 2^{-\Delta}.$$

Calculating the ratio of QT:

$$\frac{QT_{\text{QET}}}{QT_{\text{PQ}}} = \frac{(l + t \times k_{\text{QET}}) \times n \times d}{t \times n \times k_{\text{PQ}} \times d} = \frac{l}{t \times k_{\text{PQ}}} + \frac{k_{\text{QET}}}{k_{\text{PQ}}}.$$

Since $l$ and $t$ are constants and $k_{\text{PQ}}$ is relatively large, the first term is negligible:

$$\frac{l}{t \times k_{\text{PQ}}} \approx 0.$$

Thus,

$$\frac{QT_{\text{QET}}}{QT_{\text{PQ}}} \approx \frac{k_{\text{QET}}}{k_{\text{PQ}}} = 2^{-\Delta}.$$

Therefore, our QT is reduced by a factor of $2^{\Delta}$ compared to PQ.

$\square$

The reordering time complexity in QET is $O(l \times n \times d)$. Since $l$ is a small constant, the swapping time is negligible compared to the clustering time and can be ignored.

### A.4  THE PROOF OF THEOREM 3 (TIME COMPLEXITY FOR DEQUANTIZATION)

**Theorem 3.  (Time Complexity for Dequantization)** The dequantization time complexity (DQT) of our method increases compared to PQ by:

$$DQT_{\text{QET}} - DQT_{\text{PQ}} = O(l \times n \times d),$$

where $DQT_{\text{QET}}$ and $DQT_{\text{PQ}}$ represent the dequantization times of our method and PQ, respectively.

*Proof.* In PQ, the dequantization process involves a direct mapping from indices to centroids, yielding a time complexity of $DQT_{\text{PQ}}$.

In our method, the dequantization time $DQT_{\text{QET}}$ includes the additional complexity of the inverse recursive splitting process, resulting in an increase of $O(l \times n \times d)$, where $l$ is the number of levels in the recursive splitting process, $n$ is the number of data points, and $d$ is the dimensionality.

Thus, the dequantization time complexity increases by:

$$DQT_{\text{QET}} - DQT_{\text{PQ}} = O(l \times n \times d). \tag{3}$$

$\square$

### A.5  EXPERIMENTAL RESULTS OF QUANTIZATION TIME (QT) AND DEQUANTIZATION TIME (DQT)

In this experiments, we evaluated the performance of the quantization process by measuring both Quantization Time (QT) and Dequantization Time (DQT) across multiple datasets, as shown in Figures 4. These metrics provide a comprehensive understanding of the computational efficiency of the algorithm at different compression ratios, ranging from 2 to 16.

For QT, the QET algorithm exhibits longer quantization times at lower compression ratios, but as the compression ratio increases, QT decreases significantly. This phenomenon is expected, as fewer centroids simplify the clustering process, resulting in faster compression as the compression ratio increases. Vanilla consistently achieves lower QT across all datasets compared to QET, making it the faster option, albeit with a slight loss in accuracy. In contrast, PQ and other baseline methods demonstrate moderate QT performance, although OPQ and LOPQ show slower performance at larger matrix sizes due to their complexity.

As for DQT, QET maintains a relatively stable performance across all compression ratios and datasets, with marginal increases in time as the compression ratio grows. Vanilla follows a similar trend but achieves faster dequantization. PQ remains competitive in terms of both QT and DQT.

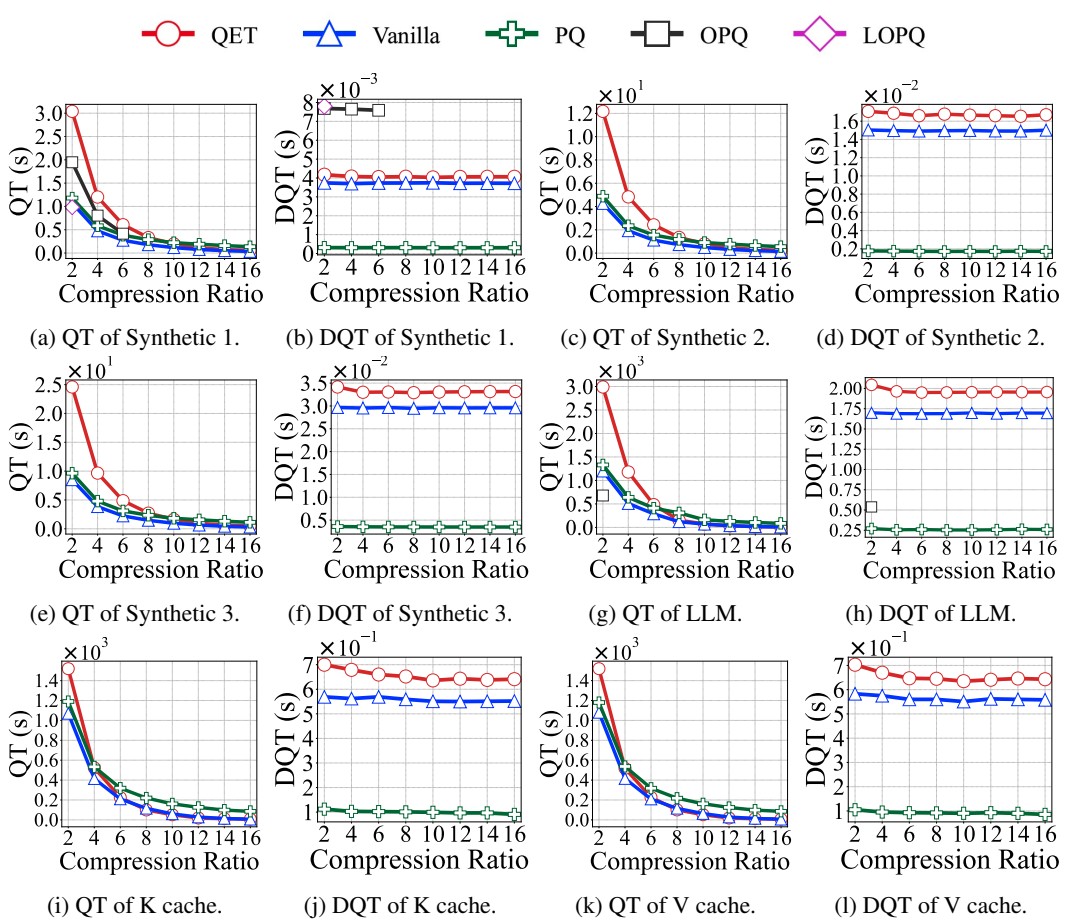

Figure 4: QT (s) and DQT (s) of different datasets.

