# OpenReview forum: "Quantum Entanglement Trees: Optimizing Quantized Matrix Quantization via Element Replacement and Residual Clustering"
_ICLR.cc/2025/Conference — ICLR 2025 Conference Withdrawn Submission_

### Official Review · Reviewer_3PFM · 2024-10-30

**Soundness:** 2
**Presentation:** 1
**Contribution:** 2
**Rating:** 3
**Confidence:** 5

**Summary:**

This paper introduces a way to compress sets of vectors (weights of linear layers or token embeddings).
The method relies on a re-ordering of vector dimensions prior to applying product quantization, possibly followed by vector quantization.
The authors demonstrate that the method improves the accuracy compared to other PQ variants.

**Strengths:**

S1. The paper takes into account the size of the codebooks in their evaluation, and proposes to compress the codebooks as well (using scalar quantization)

S2. The method can be seen as a way to replace the quantization of vector components with the compression of permutation matrices.

**Weaknesses:**

W1. The  explanation of the method is imprecise and intuitive to a point that is impossible to follow. Section 3.1 does not describe in which dimension the "elements" are compared, the mulitple matrices are not introduced, the "regularity of matrix elements" is not defined. Section 3.2 re-states the algorithm in a way that is not much more intelligible -- what does "based on their sizes" mean? In algo 2, lines 5-13 why is i divided by 2, line 16 is a for loop, lines 18-19 are hand-waivy. In the "theorems" \Delta is not introduced.

W2. There is no justification of why the method would be working. Shuffling the vector dimensions independently means that elements of different dimensions are used to train the clustering, destroying the data distribution of a given PQ subvector

W3. The comparison with the SOTA uses relatively weak baselines -- PQ variants without residuals, while it is well known that quantization with residuals works better.

W4. The comparison is in terms of MSE and MAE, not end metrics. For example, for the KV cache compression the resulting attention matrix could be compared in MSE with the exact attention result -- and ideally be tried end-to-end. The text analysis is not much more clear.

**Questions:**

There is no way that this paper can be revised to be published. It should be rewritten entirely.

---

### Official Review · Reviewer_G9Qw · 2024-11-04

**Soundness:** 2
**Presentation:** 2
**Contribution:** 2
**Rating:** 3
**Confidence:** 4

**Summary:**

The paper proposes a new method for matrix quantization based on improved local ordering of the matrix elements. The method is based on recursively rearranging adjacent elements of the matrix, and then applying product quantization to the resulting matrix.

**Strengths:**

- Matrix quantization is an important practical problem with many uses in modern applications of machine learning, such as quantizing LLM weights.
- The proposed method is conceptually simple and is easy to implement in practice. The method achieves a better compression ratio than PQ with fixed memory usage.

**Weaknesses:**

I do not see what the purpose of proposing the Quantization Error Minimization (QEM) problem is since it just adds an unnecessary penalty term to the obvious problem formulation of minimizing the MSE subject to a memory constraint. It would be helpful for the authors to clarify this point. The specific QEM formulation with the penalty term is not used anywhere, including in the derivation of the QET algorithm or any of the experiments.

The proposed QET algorithm is based on a local ordering of the matrix elements before applying product quantization (PQ). Naturally, this increases both the dequantization and the quantization time compared to PQ, but this is not evident in the main paper (you can see it in Figure 4 in the Appendix). Theorem 2 (which states that the quantization time decreases) essentially states that for a given memory usage, you can afford to use fewer centroids, and therefore the clustering will be faster. This is misleading: you should state complexities depending on the number of centroids -- if your method can achieve the same compression with fewer centroids, that is a separate point. More importantly, Theorem 3 (which states that dequantization time increases) as well as experiments on the (de)quantization time are hidden in the Appendix.

Essentially, by spending more time on both quantization and dequantization, you are able to achieve better compression. However, most applications, including the motivating applications mentioned in the paper such as LLM weight quantization and vector databases, care greatly about the dequantization time. If you care about pure compression, you could afford to run a more comprehensive optimization method. The proposed method is much slower for dequantization and is also less suitable for implementation on hardware accelerators. Finally, the two proposed tweaks over vanilla QET are very straightforward and you can also apply them to PQ.

In the experiments, you should not compare against OPQ and LOPQ as they are not really suitable for this problem. Instead, you should consider comparing against e.g. the method proposed in the SqueezeLLM paper. It would also be ideal to demonstrate that your method is suitable for the problems you use as motivation, such as LLM weight quantization, in practice. However, what you'll likely find is that since e.g. the codebook should fit into the L1 cache on GPUs, methods based on PQ are not very suitable for the problem (which is why you don't see them used in practice). Finally, you should include the experiments on the dequantization time in the main paper.

To improve the presentation, you should present the novel components of the algorithm more clearly by writing the description of Algorithm 1 (and its pseudocode) such that it refers to PQ where appropriate or uses it as a subroutine. You should also include a related work section where you discuss the relationship of your method to the prior work more clearly. Further minor points on presentation:
  - $\Delta$ in Theorem 2 is only explained in the Appendix.
  - It is confusing that LOPQ is in the legend of Figure 2 but does not appear in any of the figures.
  - Use \citep instead of \cite for proper parenthetical citations.
  - I would reconsider the name of the method, as the link to quantum entanglement is tenuous.

**Questions:**

- What is the purpose of including both the penalty term and the hard constraint in the formulation of the QEM problem?
- Which method is used for the RTN algorithm (e.g. absmax quantization)?
- Please check your code release, currently when running your provided QET example as-is, the following error occurs: ValueError: Cannot ask kmeans2 for 0 clusters (k was 0).

---

### Official Review · Reviewer_23C4 · 2024-11-11

**Soundness:** 2
**Presentation:** 2
**Contribution:** 2
**Rating:** 5
**Confidence:** 2

**Summary:**

This paper focuses on matrix quantization problem. The authors firstly formulate Quantization Error Minimization (QEM) problem, and then propose Quantum Entanglement Trees (QET) to address the QEM problem, with two additional optimizations. Experiments show the effectiveness of the proposed method.

**Strengths:**

1. This paper studies an important problem, i.e., matrix quantization, since reducing memory consumption of LLM contributes the saving of resources.
2.  The authors give theoretical proofs about lower loss and time complexity.
3. The experiments show a consistent gains over baselines.

**Weaknesses:**

1. The quality of presentation is low. Firstly, the first two paragraphs of Introduction and whole Conclusion are highly repeated in Abstract. Secondly, the authors did not include a section about related work, so I cannot fairly evaluate the contribution of this paper over current developments.
2. Memory of Indicator Map. This paper attempts to force local orderliness, but this operation will need additional spaces to save Indicator Map. Specifically, the additional space at each layer is always half of original matrix. In this case, although the matrix is quantized finally, these spaces are not ignorable.
3. Experimental baselines are too old, where the newest one was published in 2014. Honestly, this direction is not my major, so I do not know if there are other related work to compare published recently.
4. The Residual Quantization Optimization essentially does the quantization-dequantization process twice, doubling the algorithm complexity and accumulating the errors.

**Questions:**

1. What's the clustering algorithm used in this method? What's the reason for such choice?
2. Typo. At the end of Line 199, 'as shown in Algorithm 2' -> 'as shown in Algorithm 1'.

---

### Official Review · Reviewer_jbmh · 2024-11-12

**Soundness:** 3
**Presentation:** 3
**Contribution:** 1
**Rating:** 5
**Confidence:** 3

**Summary:**

The paper proposes a method to compress matrices by inducing local order (via swapping) amongst matrix elements using an iterative technique and then leveraging known product quantization techniques to compress the matrix. They further quantize the codebook and residual matrix to reduce their space requirements.

**Strengths:**

- The paper is well written and wasy to understand.
- The central idea is a simple, effective and intuitive technique to increase the efficacy of current quantization methods.
- They showcase their method on various modern data domains of interest (datasets, LLM layers and KV caches) which are all active and pertinent areas of research.
- Their method outperforms sensible baselines and works on cases whether other methods fail to work (square matrices compressed using OPQ etc)

**Weaknesses:**

# Major
- I am concerned about the technical novelty since the notion of element re-ordering for better compression is well known and has been suggested in various works [1-2]. I am sure there are more references that can be found on doing a deeper survey.
- Moreover, the key contributions which are "Abstracting the problem", "Designing the QET algorithm" are reasonable expectations of any technical work.
- "Additional Optimizations" are quite straightforward applications of RTN quantization on the artifacts involved in most compression techniques.
It is my opinion that the differential addition of the work compared to known concepts and literature does not meet the bar to publish.

# Minor
- The name of the technique is quite unrelated to the content of the paper. Even the notion of trees is simply due to the construction of recursive partitioning to the best of my understanding.
# References
1 Olken, Frank, and Doron Rotem. "Rearranging data to maximize the efficiency of compression." Proceedings of the fifth ACM SIGACT-SIGMOD symposium on Principles of database systems. 1985.
2 Chhugani, Jatin, et al. "Compressing large boolean matrices using reordering techniques." Proceedings 2004 VLDB Conference: The 30th International Conference on Very Large Databases (VLDB). Morgan Kaufmann, 2004.

# Typos
- Missing space on L139

**Questions:**

I am a bit confused about the overarching intent when limiting the occupied space to the original size of the matrix "under the condition that the quantized matrix occupies the same memory space". If the intent is to compress the matrix, one would believe that the output artifacts (quantized data + codebook + residuals) should occupy less space than the original in order to justify it being compression. The only other reason to compress would be for computational running time benefits which is not the metric being optimized in the paper.

---

### Note · Authors · 2024-11-17

I have read and agree with the venue's withdrawal policy on behalf of myself and my co-authors.